# Edge-assisted Real-time Dynamic 3D Point Cloud Rendering for Multi-party Mobile Virtual Reality

## ABSTRACT

Multi-party Mobile Virtual Reality (MMVR) enables multiple mobile users to share virtual scenes for immersive multimedia experience in scenarios such as gaming, social interaction, and industrial mission collaboration. Dynamic 3D Point Cloud (DPCL) is an emerging representation form of MMVR that can be consumed as a free-viewpoint video with 6 degrees of freedom. Given that it is challenging to render DPCL at a satisfying frame rate with limited on-device resources, offloading rendering tasks to edge servers is recognized as a practical solution. However, repeated loading of DPCL scenes with a substantial amount of metadata introduces a significant redundancy overhead that cannot be overlooked when enabling multiple edge servers to support the rendering requirements of user groups. In this paper, we design *PoClVR*, an edge-assisted DPCL rendering system for MMVR applications, which breaks down the rendering process of the complete dynamic scene into multiple rendering tasks of individual dynamic objects. PoClVR significantly reduces the repetitive loading overhead of DPCL scenes on edge servers and periodically adjusts the rendering task allocation for edge servers during the application running to accommodate rendering requirements. We deploy PoClVR based on a real-world implementation and the experimental evaluation results show that PoClVR can reduce GPU utilization by up to 15.1% and increase rendering frame rate by up to 34.6% compared to other baselines while ensuring that the image quality viewed by the user is virtually unchanged.

## CCS CONCEPTS

• **Information systems → Multimedia information systems**; • **Computing methodologies → Virtual reality**.

## KEYWORDS

Point cloud rendering, Multi-party mobile virtual reality, Cooperative rendering, Mobile edge computing

## 1 INTRODUCTION

Multi-party Mobile Virtual Reality (MMVR) is a technology that allows users to experience and share virtual reality scenes using portable mobile devices, which is an immersive interaction form between the human and computer with tremendous potential. With the increasing demand for social interactions among users, MMVR

**Unpublished working draft. Not for distribution.**

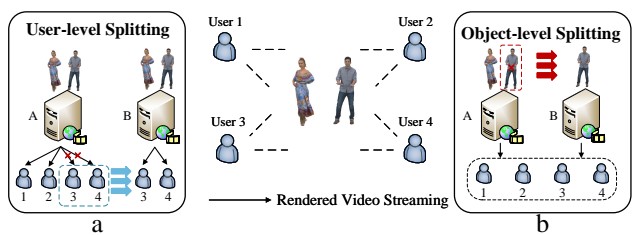

**Figure 1. Two offloading schemes of the edge rendering pipeline: a) each edge server renders the whole dynamic 3D point cloud scene for a part of users and b) each edge server renders a part of the scene for all users.**

emerged and swiftly secured its position in the market. MMVR has shown great market potential in the $11.97 billion VR market [6], such as the products of Zero Latency [4], SpringboardVR [2], and VRstudios [3]. However, the traditional content representation of MMVR can not fully meet the user requirements for higher quality and greater freedom of immersive multimedia experience. In this evolution, Dynamic 3D Point Cloud (DPCL), an emerging form of MMVR representation, can provide a free-viewpoint video with 6 degrees of freedom to bring users a more immersive experience than 360-degree video.

Rendering dynamic 3D point cloud content in real-time requires high-performance computing resources, placing a barrier to display it on constrained mobile devices. The mobile edge computing paradigm provides an intuitive solution to offload rendering tasks to edge servers equipped with graphics processing capabilities near users [13, 31]. Given that a single edge server struggles to meet the concurrent multi-user rendering demands of MMVR applications, it is often essential to adopt a distributed server scheme to satisfy the multi-user rendering demands.

As shown in Fig. 1 (a), a traditional method, called the user-level splitting method, partitions user rendering tasks among multiple servers to enhance the system's capacity for supporting a larger number of simultaneous users. However, rendering DPCL scenes brings new challenges to the edge-assisted rendering service because DPCL has a larger amount of data than traditional multimedia content (e.g., the captured point cloud models for VR content contain more than 100 thousand points, or even 1 million vertices [9]). Specifically, the above extension scheme requires the loading and processing of complete VR content metadata on each server, which results in redundant operations that cannot be ignored. This observation inspires us to extend the MMVR system by splitting the DPCL scene instead of the user group. As shown in Fig. 1 (b), we advocate spreading the DPCL objects across multiple rendering servers, called the object-level splitting method. Each user takes the rendering results from one or more rendering servers on demand and combines them into an expected video stream. This splitting mode can greatly reduce the computation overhead in edge servers

so that the MMVR system can provide higher-quality services of rendering for more users.

Realizing the object-level splitting method for edge-assisted rendering requires us to overcome two key challenges. First, the rendering results provided by the edge server are generally transmitted directly to a user in the form of a video stream while the user can not merge them directly. This is because the rendering operation reduces the data dimension from the three dimensions of a DPCL frame to the two dimensions of a normal video frame. The direct overlap of two video frames lacking 3D spatial information may result in content display errors, seriously affecting the user Quality of Experience (QoE). Second, the compatibility with the user-level splitting method needs to be carefully considered, because it is only appropriate to use the object-level splitting method when the redundancy overhead of loading is significant. We should choose the appropriate splitting method based on the computation resource demand of loading and rendering DPCL objects affected by the heterogeneous computation resources of edge servers and the rendering requirements of users.

To address the above challenges, we design a Point Cloud VR rendering system PoClVR, an edge-assisted DPCL rendering system for MMVR applications, which adds the object-level splitting method. To generate high-quality rendering results, we extract additional spatial features when edge servers are rendering and design a video blender in clients to merge the rendering result from multiple edge servers. To adapt to the heterogeneity of edge resources and dynamic rendering requirements, we design a task scheduler that converts a single-step decision into a multi-step decision process and devises an efficient heuristic algorithm to assign rendering tasks, which minimizes resource consumption.

We deploy PoClVR in a real-world implementation consisting of the controlling server, the rendering servers, and the clients. In this MMVR system, users can view a VR scene from different angles at the same time, and we evaluate the rendering performance and system resource utilization of PoClVR under various user requirements. The results show that PoClVR can improve rendering frame rate by up to 34.6% and reduce GPU resource usage by up to 15.1% compared to baseline methods while ensuring video quality. We summarize the main contributions of this paper as follows.

- A lightweight collaborative blending algorithm including 3D information extraction and rendering result blending solves a part of the visual errors caused by the object-level splitting approach.
- A dynamic task scheduling scheme effectively that adaptively optimizes the task allocation to minimize the system resource utilization while ensuring the user QoE.
- An edge-assisted cooperative rendering system for MMVR applications based on a real-world implementation, named PoClVR, and a practical evaluation of the performance and overhead achieved by an object-level splitting approach.

## 2 RELATED WORK

A lot of existing works have implemented virtual reality applications on mobile devices using edge-assisted rendering. Several works study how to use edge-assisted rendering to improve the performance of mobile augmented reality applications from the

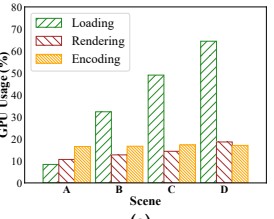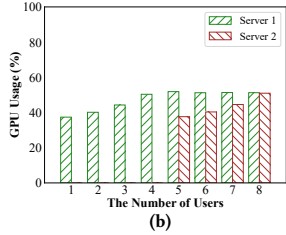

(a)             (b)

Figure 2. A case study for the resource consumption of the remote rendering by using user-level splitting mode. a) the GPU utilization of each stage in the remote rendering pipeline in 4 different VR scenes (A: longdress, B: A+loot, C: B+redandblack, D: C+soldier, each object is from *8iVFB v2* [9]); b) the GPU usage of server 1 and server 2 in scene D while the user number increases.

perspective of system design [16, 20, 21, 23]. Furion [15] designs a complete phone/server cooperative rendering pipeline that significantly reduces the latency of rendering VR scenes in real time. CloudVR [14] focuses on the interactive MVR application and deploys a cloud-accelerated MVR prototype system. RealVR [29] considers the capture and transmission of a VR scene, which only renders the scene in the user's field of view at the edge side. MoVR [5] and LTE-VR [22] optimize latency from a network perspective to meet the needs of MVR applications. Besides, some works [7, 11, 26, 27] focus on improving the transmission efficiency of panoramic video to support MVR applications. However, the above studies do not consider a rendering system for multi-party sharing scenes.

Most of the existing works on remote rendering for multi-party sharing scenes focus on how to solve the resource allocation problem to achieve the best overall QoE [10, 25]. [8, 17, 31] focuses on the resource allocation problem and proposes their iterative algorithm to solve it, respectively. [30] considers the overall object placement problem in multi-edge scenes and the rendering level selection problem for each user. Note that these work by default to deploy the rendering service to multiple edge servers using user-level splitting as shown in Fig. 1 (a). In addition, [28] considers the load conflict between the rendering task and other tasks in the MMVR scenario. However, the above studies fail to consider the impact of the redundant operations introduced by the user-level splitting method on system performance.

## 3 MOTIVATION

The fundamental reason we should introduce a new cooperative rendering method for edge-assisted MMVR systems is that the unbearable computational resources occupied by edge servers to load and manage the metadata of VR contents as VR scenes become more complex. In this section, we elaborate on this point by conducting a motivational study based on a real-world implementation to illustrate this overhead.

We deployed an edge-assisted collaborative rendering system with C++ and OpenGL [1], a foundational component for high-performance graphics, into two servers (called server 1 and server 2) with the same configuration (NVIDIA TESLA T4 GPU, 2.5G Hz Intel Xeon Platinum Skylake CPU, 15GB memory), which renders the VR scene composed of the DPCL objects from *8iVFB v2* [9]. We use *glBufferSubData* in OpenGL API to achieve real-time loading.

Figure 3. The system design of PoClVR as an edge-assisted cooperative rendering system for MMAR applications.

The rendering operation is achieved by calling *glDrawElements* to perform perspective projection and rasterization. The system prioritizes server 1 to serve the users. If server 1 is incapable of handling all rendering services, server 2 is activated to extend the system capacity as illustrated in Fig. 1 (a). We measured the correlation between GPU utilization and the number of users when varying the load of DPCL objects and plotted the result in Fig. 2.

The results illustrate two key issues. First, a single rendering server has resource bottlenecks. Therefore, we have to use a splitting scheme to split the rendering task of a user group into multiple subtasks and place them on multiple edge servers. Second, the additional cost of enabling a new server (when the 5th user joins) is greater than that of only adding a new user. This is because enabling a new server requires repeated loading of the same VR scenes. GPU resources are significantly wasted due to the redundant loading of scenes, which is evidently not the most optimal decision. This observation inspired us to decompose the VR scene by rendering a partial VR scene on each rendering server, as Fig. 1 (b) shows. This approach will reduce the waste of resources caused by redundant operations so that the overall system capacity will increase.

## 4 SYSTEM DESIGN

### 4.1 Overview

To address the above challenges, we propose PoClVR, an edge-assisted cooperative rendering system for MMVR applications, as shown in Fig. 3, including the controlling server, the rendering servers, and the clients. The controlling server is mainly responsible for controlling the logic of the MMVR application and responding to user interaction requests. When the user group makes a request, the controlling server offloads the rendering task of each required DPCL object to one or more rendering servers based on the decision of the task scheduler. The rendering server is located on an edge server, which is responsible for providing the user with the rendering results and feature information. The client renders the video

streaming received from the rendering server to the user. Note that when enabling multiple rendering servers to provide rendering services, the client should blend their rendering results by the video blender to ensure that the user can watch the right content.

PoClVR has three key components: the task scheduler on the controlling server, the feature extractor on the rendering servers, and the video blender on the clients. The task scheduler makes an appropriate task offloading decision based on benchmark rendering tests of the target object and monitoring of server status when the user group joins. Based on this decision, the clients establish one or more connections with their rendering servers decided by the controlling server. After offloading the rendering task, each rendering server will continuously generate a video stream based on the current rendering mode. The feature extractor extracts the spatial features when using the object-level splitting mode and combines these features with the video frame to create a data block. The rendering servers deliver the data block to the clients as a rendering result. The video blender in the clients blends the rendering results of multiple rendering servers. Note that the feature extractor and video blender are only enabled when using the object-level splitting mode. Otherwise, the rendering servers only deliver the video frame as the rendering result and the clients only need to decode and show the video frame. In these components, the feature extractor and video blender achieve cooperative rendering based on object-level splitting and the task scheduler ensures the compatibility of PoClVR for simple scenarios.

### 4.2 Object-level Splitting for Cooperative Rendering

The feature extractor and video blender are two key components for achieving object-level splitting. The key challenge of cooperative rendering based on object-level splitting lies in accurately representing the relationships (like occlusion, shading, etc.) between objects on the client. In this paper, we focus on ensuring the occlusion relationship between DPCL objects is correct under perspective

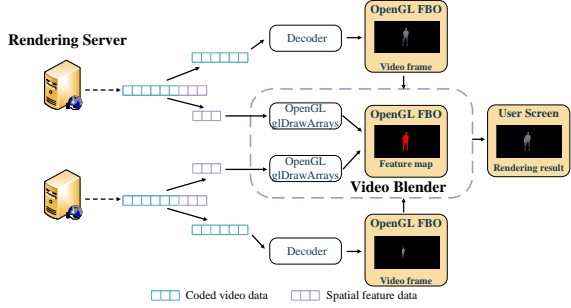

**Figure 4. The video blender draws a fusion feature map based on the spatial features generated by multiple rendering servers into the frame buffer object (FBO) using OpenGL functions. The rendering result shown on the user screen is drawn by merging video frames based on the fusion feature map.**

projection, as it is integral to all 3D renderings. To address this challenge, we propose an efficient depth feature extracting algorithm for the feature extractor to extract the spatial features during rendering and design an OpenGL-based video blender to quickly blend video frames on the client.

*4.2.1 Feature Extractor.* The feature extractor needs an extracting algorithm to serve the purpose of extracting relevant information to determine the appearance of the current object within the final scene. An intuitive approach is to deliver a compressed depth map as a spatial feature. This idea is inspired by rendering multiple objects on the same server, where the rendering engine can affirm occlusion relationships between different objects based on depth-buffering techniques [18]. There are two types of compression methods for depth maps: lossless and lossy compression. The compression ratio of lossless methods [24] is only about 4x, in contrast, the size of the compressed video frame will be much smaller than the compressed depth map, which is too costly from the perspective of bandwidth occupation. Lossy compression methods, such as the HEVC-based method [19], can lead to sharp edges of the decoded depth image, making it impossible to mix video frames of multiple objects properly. Therefore, sending the compressed depth map as a spatial feature is not feasible due to the inadequacy of existing compression methods for meeting our requirements. Another intuitive method is to deliver the average depth value of the depth map within the user's field of view. This method may overlook many details due to the irregularity of object shapes and the uncertainty of the user's perspective. We attempt to adopt a compromise solution that preserves the original 3D features as much as possible while ignoring some depth details.

To address the challenge of extracting the spatial information, we propose a feature extraction algorithm (as shown in Algorithm 1) for PoClVR capable of representing occlusion relationships in collaborative rendering with acceptable overheads. This algorithm traverses the depth map in line scanning mode and merges contiguous pixels with similar depth values into one data segment. Each data segment is recorded with its starting and ending coordinates, along with the average depth of pixels within that segment. The edge server extracts all the depth segment information after rendering the video frame and sends it to the client as the spatial feature.

---

**Algorithm 1:** Depth Feature Extracting

---

1 **Input:** A $w * h$ depth map of current video frame.
2 **Output:** A sequence $P$ can be sent.
3 Create empty queue $Q$ to collect triples $(l_1, l_2, d)$;
4 Initialize $l_0 = -1, step = 4$;
5 **for** *Line i in* $\{0, 1, \ldots, h - 1\}$ **do**
6     Set column $j = 0$ and $d_{sum} = 0$;
7     **while** $j < w - 1$ **do**
8         Calculate the location of these pixels $l = i * w + j$;
9         Record the depth value $d_{sum} += d_{i,j}$;
10         **if** $l_0 == -1$ *and* $d_{i,j} \neq 1.0$ **then**
11             Find the start location $l_0$ between $l - step$ and $l$;
12             Record depth value $d_{sum} += \sum_{k \in [l_0, l)} d_{i,k}$;
13             Push $l_0$ to $Q$;
14             Set $step = 1$;
15         **else if** $abs(d_{i,j} - d_0) > threshold$ **then**
16             Push $l - 1$ to $Q$;
17             Push average depth value $d_v = d_{sum}/(l - l_0)$ to $Q$;
18             Reset $l_0 = -1, step = 4$;
19         **end**
20         $j = j + step$;
21     **end**
22     **if** $l_0 \neq -1$ **then**
23         Push $l = (i + 1) * w$ to $Q$;
24         Push average depth value $d_v = d_{sum}/(l - l_0)$ to $Q$;
25         Reset $l_0 = -1$;
26     **end**
27 **end**
28 Converts the queue $Q$ into the byte sequence $P$;

---

This extracting algorithm effectively reduces data volume through the following observation: the point cloud object to be rendered always contains various continuous surfaces, and these surfaces often have sections with similar depth values regardless of the camera perspective. Thus, to determine the correct occlusion relationship, the client can assess the occlusion among these sections using the average pixel value of them, without requiring the precise depth value of each pixel.

*4.2.2 Video Blender.* The workflow of client-side video blending is shown in Fig. 4. The client receives data blocks from the rendering servers and divides each block into video data and feature data. The video data can be quickly decoded as an image by a decoder. The image will be stored in an OpenGL frame buffer object (FBO) temporarily. At the same time, the video blender plots the spatial features as a fusion feature map in an FBO by calling *glDrawArray* function. After features have been plotted into the feature FBO, the video blender blends the color value of each pixel based on the fusion feature map. The benefit of this feature blending method based on OpenGL drawing operations is that it can fully utilize the parallel processing capabilities of mobile GPU. Moreover, the results obtained can be directly used in subsequent drawing operations, greatly accelerating the speed of video blending. Note that the fusion feature map should first be plotted into the FBO whose

resolution matches that of raw video, and then scaled into a texture with the resolution of the user screen. Due to the line scanning of the extract algorithm, if the FBO resolution during rendering is smaller than the resolution of the user screen, the fusion feature map will appear as empty lines.

## 4.3 Task Scheduling

Note that not all scenarios are suitable to utilize the object-level mode. The primary duty of the scheduler in PoClVR is to determine the rendering mode and make the adaptive task offloading decision based on the task requirements and the remaining amount of computing resources. Consequently, the scheduler must ascertain which mode is more suitable for the present user requirements. If using object-level splitting mode, the scheduler should adjust the rendering tasks allocation based on the changes in the server state and user requirement. It performs the scheduling algorithm when a new user group enters and periodically adjusts the decision to deal with possible environmental changes.

*4.3.1 Task Allocation Problem.* Before designing the scheduling algorithm, we first study the task allocation problem in the MMVR system which contains a group of users with the common objects and multiple edge servers. We denote the user set and the edge rendering server set as $\mathcal{U}$ and $\mathcal{R}$, respectively. The users request a rendering service including multiple tasks denoted as the set $\mathcal{T}$. The first decision variable is rendering mode $m$, where $m = 0$ means choosing the user-level splitting mode and $m = 1$ means choosing the object-level splitting mode. The second decision variable is the task offloading matrix, which is defined as $\mathcal{X} = \{x_r^{(t)} \mid r \in \mathcal{R}, t \in \mathcal{T}\}$, where $x_r^{(t)} = 1$ means the rendering task $t$ is offloaded to the rendering server $r$. Besides, since the user connecting additional servers during the application running may cause unacceptable delays, we arbitrate a valid server set $\mathcal{R}' \subseteq \mathcal{R}$ during the initialization phase, as the third decision, which ensures a smooth experience.

To characterize the computational resource consumption (mainly GPU resources for rendering tasks) in PoClVR, we formulate the total GPU occupation time for rendering server $r$, which includes rendering time, loading time, encoding time, and additional overhead for object-level splitting. Specifically, we use $c_r^{(t)}$ to denote the GPU occupation time for rendering task $t$ in server $r$, which depends on the target object in task $t$ and the resources of rendering server $r$. Based on the real-world evaluation, we believe that both the rendering time and the loading time are relevant to the vertex count of DPCL objects. Therefore we apply the fitting function $f_r(\cdot)$ for rendering server $r$ and the vertex count $v_t$ of task $t$ to calculate the rendering time $c_r^{(t)} = f_r(v_t)$. Similar to function $f_r(\cdot)$, we use $l_r(\cdot)$ to denote the fitting function of real-time loading time in rendering server $r$. By using $N = |\mathcal{U}|$ to denote the number of users in the current user group, the total rendering time and loading time for server $r$ can be calculated as follows:

$$e_r^{rl} = \sum_{t \in \mathcal{T}} \left(N \cdot c_r^{(t)} + l_r(v_t)\right) \cdot x_r^{(t)}, \forall r \in \mathcal{R}. \quad (1)$$

We further use $e^e$ to denote the time of encoding a video frame by GPU for each user, thus the total GPU occupation time $e_r$ for $r \in \mathcal{R}$

can be calculated as follows:

$$e_r = e_r^{rl} + N \cdot e^e + N \cdot e_r^o \cdot \mathbb{I}(0 < \sum_{t' \in \mathcal{T}} x_r^{(t')} < |\mathcal{T}|), \quad (2)$$

where $\mathbb{I}(\cdot)$ is the indicator function and $e_r^o$ represents the additional overhead required by object-level splitting. Therefore, we can now estimate the total resource consumption $C$ as follows:

$$C = \sum_{r \in \mathcal{R}} \frac{e_r}{E}. \quad (3)$$

where $E$ denotes the sampling time of a single frame (e.g., 1/30s for a 30-fps video).

As for the performance metric, we use the number of tasks running on a render that exceeds the performance bottleneck, which is denoted as $F$ and can be calculated as the following equation:

$$F = \sum_{r \in \mathcal{R}} \mathbb{I}(e_r > \epsilon_r E) \cdot \sum_{t \in \mathcal{T}} x_r^{(t)}, \quad (4)$$

where $\epsilon_r \in [0, 1]$ depends on the available GPU time for rendering on the physical machine of the server $r$. Specifically, $\epsilon_r = 1$ when the rendering process completely occupies the GPU, and $\epsilon_r$ is less than 1 when other application processes occupy part of the GPU. Whenever the rendering task is executed on an edge server that exceeds the bottleneck (i.e., $e_r > \epsilon_r E$), the FPS experiences a significant drop. Therefore, $F$ is a strongly serious penalty, which means that we have to bias this value towards 0 when making decisions.

Finally, imaging quality significantly affects the QoE of users. We use $q_u$ to represent the imaging quality of user $u$, which mainly depends on the rendering mode $m$ since object-level splitting comes with some quality loss of video frames. The average QoE of all users in the MMVR system can be calculated as follows:

$$Q = \sum_{u \in \mathcal{U}} \frac{q_u}{N}, \quad (5)$$

Based on the total resource consumption $C$, performance metric $F$ and average QoE $Q$, we formulate the optimization problem in PoClVR as follows:

$$\underset{m, \mathcal{X}, \mathcal{R}'}{\text{maximize}} \quad P(m, \mathcal{X}, \mathcal{R}') = Q - \alpha C - \beta F; \quad (6)$$

$$\text{s.t.} \quad \sum_{r \in \mathcal{R}'} x_r^{(t)} = 1, \forall t \in \mathcal{T}, m = 0; \quad (7)$$

$$x_r^{(t)} = 1, \forall r \in \mathcal{R}', \forall t \in \mathcal{T}, m = 1; \quad (8)$$

$$x_r^{(t)} \in \{0, 1\}, \forall t \in \mathcal{T}, \forall r \in \mathcal{R}, \forall m = 0, 1; \quad (9)$$

where $\alpha > 0$ and $\beta > 0$ denote the weight coefficients of the resource consumption and the performance metric. Without loss of generality, we assume that the system operates in a time-slot manner, with each slot interval corresponding to a decision period denoted by $L$. The scheduler should solve the above optimization problem at the beginning of each period and offload the rendering tasks based on the optimal decision $\mathcal{X}$.

---

**Algorithm 2:** Task Scheduling

1 **Input:** User set $\mathcal{U}$, rendering server set $\mathcal{R}$, and task set $\mathcal{T}$
2 Initialize available server set $\mathcal{R}' = \mathcal{R}$;
3 $P_0 \leftarrow$ getUserOptimal($\mathcal{R}'$);   // if m=0
4 $P_1 \leftarrow$ getObjectOptimal($\mathcal{R}'$);   // if m=1
5 Set $m = \arg\max_{m \in \{0,1\}} P_m$;
6 **if** $m = 0$ **then**      // user-level splitting mode
7    $\mathcal{R}' \leftarrow$ solveUserMode();
8    Offloading all user tasks into server $r \in \mathcal{R}'$;
9 **else**      // object-level splitting mode
10    **while** $Time\ interval \geq L$ **do**
11      $\mathcal{X} \leftarrow$ solveObjectMode();
12      Set $\mathcal{R}' = \{r | x_r^{(t)} = 1, \forall t \in \mathcal{T}\}$ (only on the first loop);
13      Offloading the tasks based on $\mathcal{X}$;
14    **end**
15    Monitor all rendering servers and update the remaining available resources $E - e_r, \forall r \in \mathcal{R}'$;
16 **end**

---

*4.3.2 Estimation and Algorithm.* Since the dynamic system information, such as the fitting functions $f_r(\cdot)$, $l_r(\cdot)$ and the parameters $d^e$ and $d_r^o$, are difficult to accurately predict in each decision interval, we estimate initial values by performing benchmark rendering tests for each rendering target in advance. Benchmark tests can be done when preparing the DPCL contents so that users do not need to wait unnecessarily with the system operating. We first render each object individually with real-time loading, and the camera faces the object and circles around it within 5s. Then, the rendered results are encoded into video streams by GPU, and we record the GPU running time during this process, which can be an estimation for $e_r$. Given the heterogeneous nature of the rendering servers, we can benchmark them on every available type of GPU across all the rendering servers.

Relying solely on the initial values to make decisions is insufficient for adapting to a dynamic environment and rendering requirements. Therefore, the scheduler must monitor and update these estimates throughout the rendering process. To achieve this, we set a 2-second decision period (i.e., $L = 2$) and employ a moving average method for updating the estimates. The scheduler needs to make a decision as soon as possible at the beginning of each period but directly solving the optimal solution of the above optimization problems is difficult. To address this challenge, we propose an efficient heuristic algorithm as shown in Algorithm 2.

In the initial phase, we set $\mathcal{R}'$ to $\mathcal{R}$ and determine the extended mode variable $m$. The scheduler finds the optimal decision matrix $\mathcal{X}$ for each mode $m$ ($m = 0$ or $m = 1$) and calculates the optimal objective value $P_m$ in (6) by the function getUserOptimal and getObjectOptimal respectively. The mode $m$ with the larger value of $P_m$ is finally selected as the extended mode of the current user group (line 5). Note that $m$ and $\mathcal{R}'$ cannot be changed during the subsequent decision process because switching the connection between users and servers at any stage other than initialization can devastate the user experience.

In user-level splitting mode, the scheduler only needs to select $\mathcal{R}'$ without considering the task allocation. We encapsulate the function solveUserMode, which sorts the rendering servers according to the remaining resources (i.e., $E - e_r$) and selects a rendering server $r$ for each user $u \in \mathcal{U}$ to form the set $\mathcal{R}'$. The scheduler then offloads all the user tasks into the server $r \in \mathcal{R}'$. In object-level splitting mode, we design the function solveObjectMode to solve the optimization problem (6) in each decision period. Specifically, we first transform the matrix decision $\mathcal{X}$ problem into a multi-step task sequence decision $\mathcal{X}' = \{x^{(t)} | t \in \mathcal{T}\}$, where $x^{(t)} = r$ means that task $t$ will be offloaded to server $r$. Then, we choose the server $r$ for each task $t$ in turn based on the penalty value, which can be calculated as follows:

$$p_r = \Delta Q_r + \alpha \Delta C_r + \beta \Delta F_r + \gamma \frac{C_{now}}{\epsilon_r E}, \quad (10)$$

where $\Delta Q_r, \Delta C_r, \Delta F_r$ denote the change of $Q, C, F$ affected by allocation the current task into server $r$ respectively. *gamma* $> 0$ is a weight coefficient similar to $\alpha, \beta$. Note that we introduce $C_{now}$ into this penalty function to incentivize the algorithm to prefer assigning the task to the rendering server with the most residual resources, provided that other factors are relatively similar. If the decision-making process is conducted only once per task based on this penalty value, the algorithm likely favors filling up a single rendering server before considering object-level splitting. To prevent such lopsided decisions, it is necessary to implement a secondary decision-making process during the execution of the algorithm. This involves attempting to modify one of the current task assignment decisions in the decision matrix sequentially and selecting the option with the lowest penalty value as the new preference. This method guarantees an equitable utilization of the rendering server in situations that require object-level splitting. After the first decision, we set $\mathcal{R}' = \{r | \sum_{t \in \mathcal{T}} x_r^{(t)} > 0, r \in \mathcal{R}\}$ (line 12), and operate the offloading decision $\mathcal{X}$.

## 5 PROTOTYPE IMPLEMENTATION

We design a test MMVR application to render multiple avatars of dynamic virtual objects and show them to the user. The client obtains moving and rotating view instructions from users through tactile sensors. These instructions are sent to edges in real time to update camera parameters and render new perspectives. Each client can request to be served by one or more edge servers. Note that the traditional edge-assisted rendering is used when the client is served by only one server.

As shown in Fig. 3, we implemented the edge server with 3K lines of C++ in a server with Nvidia GeForce GTX 1660 SUPER. We use OpenGL to render point cloud videos from *8iVFB v2* [9] and use Nvidia Video Codec SDK to accelerate the encoding process with Nvidia GPU. The edge server will create an EGL context for each user and draw virtual objects in turn by OpenGL in the rendering loop. The video streaming generated by the encoder will be sent by using WebSocket as the network transport protocol, which is chosen because it can actively push video streams from the server side in real time. The feature is extracted after each frame is rendered and inserted into the encoded video data block in the form of a byte stream. The client is implemented as an Android APP with 2K lines

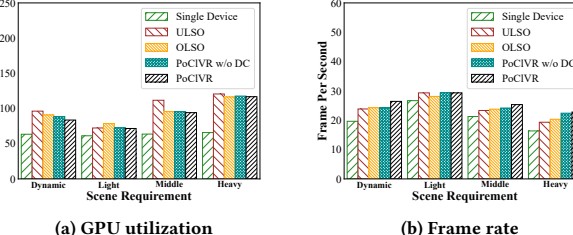

(a) GPU utilization    (b) Frame rate

Figure 5. Comparison between PoClVR and baselines on (a) GPU utilization and (b) frame rate.

of JAVA in a Redmi Note 11 Pro with the MediaTek Helio G96 chip. It establishes a WebSocket connection with each server required by it to send user interaction data and receive video streams with the feature. We use the Android Mediacodec hardware decoder to decode the video stream and store the decoded image temporarily on *SurfaceTexture*. At the same time, the feature map is drawn based on OpenGL ES. The video blender selects the pixel value from the appropriate texture for each pixel, respectively, according to the feature map, which is achieved by programming the fragment shader using OpenGL Shading Language (GLSL). Besides, we also implement the feature of rendering point cloud videos directly using raw data by OpenGL ES on the clients. This feature is used to demonstrate that it is difficult for clients to directly support the rendering of complex objects.

## 6 EVALUATION

### 6.1 Experiment Setup

We deploy the rendering server on two servers with a 16-core Intel CPU, and NVIDIA Tesla T4 GPU. The controller is only deployed in a light server with a 2-core Intel CPU to support the task scheduler and application logic. We build two implementations of the client in our experiments. 1) Simulation script: a thin client that only requests the rendering service. 2) Android App: a client is deployed in Redmi Note 11 Pro running Android 11 with MediaTek Dimensity 920 SoC. First, we simulate multiple user rendering requests using the simulation script to verify that PoClVR can reduce system compute resource usage and increase the average render frame rate compared with baseline algorithms. Second, we evaluate the performance of the video blender using the Android App to illustrate that PoClVR can achieve satisfactory video quality and playback frame rate.

We build our DPCL scene by using the DPCL object from 8iVFB v2[9] datasets, which contain 4 DPCL sequences. In each sequence, the full body is captured by 42 RGB cameras configured in 14 clusters, at 30 fps, over a 10 s period. We put these 4 virtual objects together as the VR scene playing at 30 fps and multiple users can join it at the same time. When each user joins the scene, a virtual camera is created and it rotates around this scene at a fixed rate. The content it captures is presented to the user as a video. To quantify user requirements, we randomly generate user requirements with DPCL object as the unit. Specifically, the user can choose to load any of the above four objects 0 or 1 or 2 times and arbitrarily combine the four objects, which means we have $3^4$ possible requirement.

To simulate VR scenes with different overheads of loading and rendering, we set up three scenes based on the number of objects

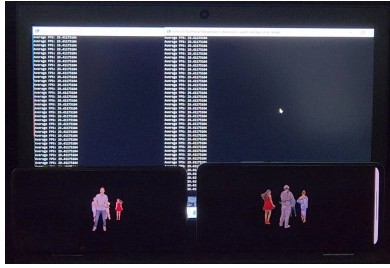

Figure 6. The PoClVR system in a real-world implementation, which consists of two rendering servers serving two clients. The DPCL scene has 4 DPCL objects, where each server renders two objects. The two clients are watching the scene from opposite sides.

the user requires: 1) light loading scene with 1-2 objects, 2) middle loading scene with 3-5 objects, and 3) heavy loading scene with 6-8 objects. This setting is empirical, as we observe that loading less than two objects is generally easy for a group of users, while loading more than six objects is hard.

### 6.2 Performance Improvement

We compare the performance of PoClVR with three baselines: 1) User-level splitting only (ULSO), which only uses the user-level splitting method to support the rendering service for the user group. 2) Object-level splitting only (OLSO), which only uses the object-level splitting method to support the rendering service for the user group. 3) PoClVR without dynamic scheduling (PoClVR w/o DC), which only allocates the objects at the system beginning.

We use *GPU usage* and *average render frame rate* as performance metrics. To simulate different user requests, we randomly generate 20 groups of different user requirements with each set comprising 5 to 10 users. Initially, we set up a rendering requirement for each user group as mentioned in section 6.1 and reset it twice subsequently to mimic shifts in user requirements. The simulated users within each group will submit their requests to the controller and maintain the rendering process for 30 seconds. Fig. 5 shows that PoClVR can reduce the GPU usage by up to 15.1% and improve the frame rate by up to 34.6% in our experiment. Note that the GPU usage when using a single edge server to render is minimal because it has no more computing resources available after it reaches the performance bottleneck. PoClVR can achieve the best performance because it dynamically considers the changing needs of the user and chooses the most appropriate rendering method.

We also compare the performance of PoClVR fixed to the three scenarios mentioned in Section 6.1 respectively. Fig. 5 illustrates the user-level splitting method is more appropriate for the light loading scene, as the overhead of the redundant loading operation is almost negligible. Conversely, the object-level splitting method is more suitable for the heavy loading scene, where repeated loading can deplete the edge server's limited computing resources. The above results verify that PoClVR can adaptively adjust the task assignment method according to different scene requirements. These experimental results prove that collaborative rendering using the object-level method in PoCLVR can guarantee almost the same image quality and acceptable latency as single-server rendering.

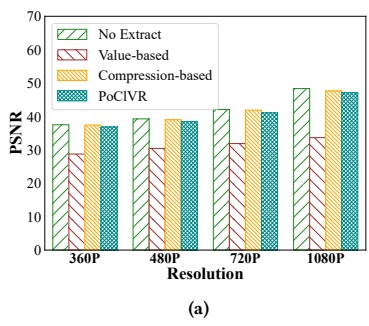 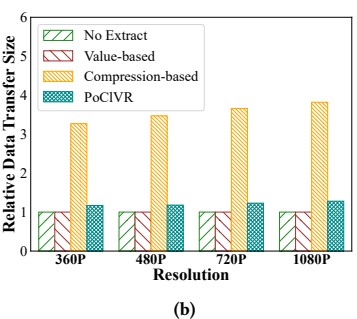 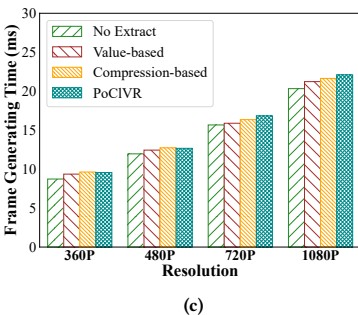

(a) (b) (c)

**Figure 7. The evaluation of our semantic video blending method on different metrics: (a) PSNR, (b) the data size to be transferred, and (c) the time of generating a frame.**

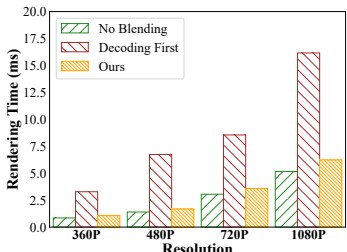

**Figure 8. Rendering time for a single frame in the client with various blending methods.**

## 6.3 User Experience Assurance

PoClVR should ensure the user experience while it improves the system efficiency. We design an experiment to verify that its collaborative rendering can guarantee a similar user experience to single-device rendering. We use two mobile phones deployed with PoClVR client to request a basic scene, whose virtual cameras capture the DPCL scene from both sides of the scene respectively. As shown in Fig. 6, we run the client and record the mobile phone screen to calculate the peak signal-to-noise ratio (PSNR) [12].

First, we compare the image quality generated by edge servers with four different depth extract methods under various video resolutions: 1) no extract, 2) value-based method, which extracts the depth as an average depth value to indicate the location of rendering results, 3) compression-based method, which uses a lossless compression algorithm (RVL [24])to compress the whole depth map, and 4) Algorithm 1. Since our feature extraction and video blending methods are significantly related to video resolution, we conducted experiments under 4 resolution configurations. Fig. 7 shows that PoClVR can provide image quality close to that of the single-device rendering with only about 0.15% difference and acceptable overheads. The value-based method requires minimal additional data to transfer, but the images shown on the user screen usually have obvious errors in the visual effects, which leads to a serious loss of image quality. The compression-based method can achieve good image quality, but it requires transferring 2.3 times the amount of data, which needs to be avoided for mobile users with costly network bandwidth resources. Fig. 7(c) illustrates the processing time of PoClVR's feature extract algorithm is only 2 milliseconds, similar to another baseline method.

**Table 1. Additional execution time of PoClVR when rendering 4 dynamic 3D point cloud objects as a 1080P video stream.**

| Processing | Execution Time (ms) | Location |
|---|---|---|
| Extracting | 3.32 | Rendering server |
| Blending | 4.56 | Client |
| Scheduling | 0.0058 | Controlling Server |

Second, we compare the execution time of video blending in the clients using three different blending schemes: 1) no blending, 2) decoding first method, which decodes the received feature to a depth map in the CPU, and 3) our method, which renders a feature map directly by GPU based on GLSL. Fig. 8 shows that the proposed blending method needs an average of 6 milliseconds to blend a frame, which is merely 0.38 times that of the decoding first method.

Third, we record the additional execution time of PoClVR when rendering 4 dynamic 3D point cloud objects as a 1080P video stream in Table 1. Since the number of servers and tasks in the scheduling algorithm is small (Algorithm 2), the overhead of the scheduling algorithm is almost negligible. The extracting algorithm (Algorithm 1) use 10.0% of a frame interval, which mainly takes up CPU time. This is acceptable in the system because the CPU is not the main performance bottleneck. The video blending time of 4.56ms is also acceptable because the client only needs to decode and blend, as long as it can be done within one frame interval.

## 7 CONCLUSION

This paper proposes PoClVR, an edge-assisted cooperative DPCL rendering system for MMVR applications, which splits complex DPCL scenes into objects and renders them using multiple edge servers. We design the feature extractor and video blender to ensure that the cooperative rendering can accurately represent the relationships between objects. To adapt to the heterogeneity of edge resources and the dynamic rendering requirements, we consider a task allocation problem and propose a heuristic algorithm to solve it. The experimental evaluation based on a real-world implementation shows that PoClVR can reduce GPU utilization by up to 15.1% and increase rendering frame rate by up to 34.6% compared to other baselines. Besides, the experiment results also show that PoClVR achieves image quality close to that of single-device rendering with only up to about 6 milliseconds of additional time.

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
