# OpenReview forum: "Edge-assisted Real-time Dynamic 3D Point Cloud Rendering for Multi-party Mobile Virtual Reality"
_acmmm.org/ACMMM/2024/Conference — MM2024 Poster_

### Official Review · Reviewer_pkqL · 2024-05-24

**Rating:** 3
**Confidence:** 3

**Summary:**

The paper describes a system that renders multiple point cloud streams for multiple simultaneous viewers, and optimises GPU usage and throughput by smartly allocating which GPU to use to render which point cloud for which user.

**Strengths:**

The idea of not assigning GPUs to users but in stead (optionally) assigning them to point cloud streams and having one GPU do the rendering of that point cloud for every user is novel, and a very smart idea.

To me, figure 2(a) is the important figure: for point cloud streams the performance bottleneck is the loading of the point cloud data into the GPU, not the rendering, and not the video capture. Therefore, if you distribute the point clouds over the GPUs and then have each GPU render/encode for multiple users your overall system throughput goes up.

**Limitations:**

The strength of the paper that I sketched above is never mentioned anywhere explicitly, it seems to be taken as a given, but it is then used everywhere else in the paper implicitly.

The description of the system is not always very clear, and sometimes even very obscure. For example the "feature extractor" is really nothing more than lossy depth image encoder, which work quite nicely for point clouds of this type (humans or probably also other shapes that are limited in space, unlike point clouds of complete environments).

Section 6.2 has an important omission: it does not explain what "Single device" means. Single end-user device or single GPU/edge server? This makes the results difficult to interpret.
In addition, the figure 7(a) and (b) are both _relative_ graphs, which makes them much less useful than if they had been absolute (because that would allow the reader to judge, say, PSNR and bandwidth usage of PoClVR at 720P to compression-based at 480P.


I'm not quite sure what the value is of the whole section on task scheduling, or actually of using anything else than OLSO. This should have become clear from section 6, but as stated above I have difficulty interpreting the data there.

**Suitability:**

2

---

### Official Review · Reviewer_5tgB · 2024-05-28

**Rating:** 4
**Confidence:** 3

**Summary:**

The paper proposes PoClVR, an edge-assisted 3D Point Cloud rendering system for multi-party mobile virtual reality. PoClVR can dynamically decide to use the user-level splitting or object-level splitting for rendering offloading to multi-edge servers. Meanwhile, a lightweight collaborative blending algorithm is designed to address the challenges raised in object-level splitting. Experiments are performed to demonstrate the efficiency of PoClVR.

**Strengths:**

•. It is novel to add object-level splitting besides user-level splitting, which can optimally select one level splitting based on factors, such as edge server resource and image quality.

•. The paper is well written and clearly state the motivation, design, and evaluation, which make it easy to follow.

•. The paper provides evaluation for the proposed scheduling algorithm and blending method.

**Limitations:**

•  It would be nice to give descriptions of the mathematical symbols in Algorithm 1 to enhance the understanding. In Section 4.3.1, the authors try to combine those two modes into a unified mathematical formulation. However, it is not so clearly described. Meanwhile, it would be nice to recheck equation (2).

• The paper is motived by GPU usage of servers while number of users increase (Figure 2). Hence, it would be beneficial to explore how the  number of users affect the performance of proposed method itself and also compared with other baselines. It is also interesting to explore the effects of environmental changes (such as number of objects changes during the continuous video period and number of users increase as new users join) on the performance of the proposed method.

• PoClVR is compared with other baselines. It is not so clear what exactly the user-level splitting only (ULSO) baseline is. As mentioned by the authors there are former research work mainly focus on user-level placement in multi-edge scenes. So would be nice in this place to compare with the latest proposed user-level splitting only (ULSO) baseline, such as [30].

**Suitability:**

2

---

### Official Review · Reviewer_fQHu · 2024-05-29

**Rating:** 2
**Confidence:** 2

**Summary:**

The paper proposes a distributed rendering method to split the workload of rendering a complex point cloud video. Instead using one server to render the whole video, it uses multiple servers to render different objects of the point cloud video and then merges the rendering results on the client for the final image. A depth extraction method is proposed to effectively compress the depth map of each partially rendered frame and a blending method is applied on the client to maintain correct occlusion.

**Strengths:**

The most novel idea of the paper is the so called feature extraction algorithm that scans all pixels line by line, finds the group of pixels that share similar depth and uses the average value to represent the whole group.

**Limitations:**

There are a couple of drawbacks of this paper:
1. Distributed rendering is actually a well studied problem in graphics. But this paper actually address a very specific scenario, the rendering of point cloud video, for which there is no lighting/shadow/reflection issue, and objects can be easily divided. In summary, the problem this paper targets is narrowed down to a very small scope.
2. The idea to cluster pixels with similar depth and use the average depth for representation might be the key of this paper, but it is not well evaluated. For example, the evaluation only compares GPU usage but not CPU usage, and the paper does mention how to implement the feature extraction in GPU. Only the extraction time (3ms) is mentioned but if it takes too much CPU for calculation, it won't scale well. Second, the size of the extracted depth map is only compared with lossless compression, while there are many other approaches trying to use lossy video codec for depth map compression.
3. The overall benefit of spliting the objects and distributing to different render servers is not fully justified. The GPU reduction numbers reported are just based on a very limited simulation test and not convincing enough. In the big picture, as long as one server can render the whole scene, I don't believe spliting objects can benefit much (if any) in terms of the overall GPU resource usage, especially considering the resources used to extract and compress depth maps. However, it may help a lot of reducing the streaming cost because we don't have to streaming all PCV streams to all rendering servers. Instead, we may just send each rendering server one single PCV stream. But the paper doesn't discuss much about this.

**Suitability:**

2

---

### Meta-Review · Area_Chair_E3FN · 2024-07-12

**Recommendation:** Accept (Poster)
**Confidence:** 3

**Metareview:**

The paper got two Borderline Accept (one upgraded from a Borderline Reject after the rebuttal) and one Weak Reject (kept after the rebuttal, and quite critical) recommendations. Thus, it was a borderline case discussed with other Area Chairs and Program Chairs (having an overall view of borderline cases), and the final recommendation is to accept this paper to be presented as a poster.